# The prevalence of mental health and addiction concerns and factors associated with depression and anxiety during the COVID-19 pandemic in Ontario, Canada: A cross-sectional study

**Oswin Chang**[1], **Anthony Levitt**[1,2,3,4], **Maida Khalid**[1], **Sugy Kodeeswaran**[2], **Roula Markoulakis**[1,5]*

**1** Sunnybrook Research Institute, Toronto, Ontario, Canada, **2** Family Navigation Project, Sunnybrook Health Sciences Centre, Toronto, Ontario, Canada, **3** Hurvitz Brain Sciences Program and Department of Psychiatry, Sunnybrook Health Sciences Centre, Toronto, Ontario, Canada, **4** Department of Psychiatry, University of Toronto, Toronto, Ontario, Canada, **5** Department of Occupational Science and Occupational Therapy, University of Toronto, Toronto, Ontario, Canada

* roula.markoulakis@sunnybrook.ca

**Data Availability Statement:** Data collected for this study, including individual participant data and a

## Abstract

### Objective

Higher than expected rates of mental health and/or addiction (MHA) concerns have been documented since the onset of the COVID-19 pandemic. A more up-to-date prevalence of MHA outcomes and the factors associated with the occurrence of MHA concerns remains unclear. This study examined the prevalence of MHA outcomes and factors associated with screening positive for symptoms of depression only, anxiety only, and both depression and anxiety two years into the COVID-19 pandemic in Ontario, Canada.

### Method

Ontario adults ≥18 years of age ($n = 5000$) reported on the presence of symptoms associated with depression, anxiety, and substance use between January and March 2022. Data were also collected on pandemic-related health variables, including COVID-19 infection fear, changes in socioeconomic status and mental health since pandemic onset, satisfaction with social supports, and MHA service needs.

### Results

The prevalence of positive screening for depressive or anxiety symptoms only was 8% and 11%, respectively, while 36% screened positive for both. Moderate/high risk levels of substance use were found in 20% of participants for tobacco and 17% for both alcohol and cannabis. Moderate/high risk levels of alcohol use and certain pandemic-related factors (negative change in mental health, unmet MHA service needs) were associated with positive screening for symptoms of depression only, anxiety only, and both depression and

data dictionary defining each field in the set, are not publicly available because they contain potentially identifying and sensitive patient information. Data can be made available upon request. Such requests should include a study protocol with clear hypotheses, which will then be reviewed by the study principal investigator and investigator team. If the hypothesis is valid and complies with the authorizations for this study, a data transfer agreement will be developed prior to transfer of de-identified data. Requests should be sent to the Sunnybrook Research Ethics Office at REB@sunnybrook.ca.

**Funding:** This work was supported by the Canadian Institutes of Health Research (AL, SK, RM; grant number UIP-178831) and the Hurvitz Brain Sciences Summer Student Research Program (OC) through the Sunnybrook Research Institute. The funders had no role in study design, data collection and analysis, decision to publish, or preparation of the manuscript.

**Competing interests:** The authors have declared that no competing interests exist.

anxiety. Satisfaction with social supports was associated with lower likelihoods of being in the depression only and both depression and anxiety groups, and non-White ethnicity was associated with depression only.

## Conclusions

There was a continued burden of MHA issues two years into the pandemic. These results underscore the ongoing need for timely and accessible MHA services.

## Introduction

Globally, depression and anxiety have been estimated to affect 4.4% and 3.6% of the population, respectively [1]. In the early months of the COVID-19 pandemic, increased rates of screening positive for depression and anxiety were well-documented in general population samples, with some prevalence estimates placing depression at 28% and anxiety at 33% [2]. These findings may be unsurprising considering the implementation of various public health measures during the pandemic, from lockdowns to social distancing, which may have upended many lives due to school closures [3], increased caregiving and household responsibilities for parents [4], healthcare services being cut [5], and greater financial difficulties [6]. However, some subsequent studies suggested that screening positive for depression and anxiety increased as the pandemic progressed [7, 8], while others suggested consistent rates or decreases [9, 10]. These heterogeneous findings may be due to varying pandemic severities in different countries, the type/length of societal restrictions, the timing of measurements, methodological differences, availability of vaccines, or the populations themselves [11–13]. Thus, a more up-to-date prevalence of these outcomes from various global jurisdictions is needed to address specific regional needs.

Various factors have been found to be associated with symptoms of depression or anxiety pre-pandemic, such as female gender, younger age, marital challenges, chronic health conditions, poor familial relationships, financial difficulties, unemployment, and family history [14–18]. During the pandemic, some studies reported experiencing symptoms of depression or anxiety was found to be related to younger age, female gender, ethnic minority status, low socioeconomic status (SES), living situation, and non-married marital status [19–22]. However, not all of these factors were found to be significant in all studies, which suggests that additional contextual influences may be an important consideration, so further exploration into people's experiences with the pandemic itself is warranted. Considering factors such as access to healthcare services, social support, or fear are related to mental health and have been directly impacted due to the pandemic or associated public health measures [5, 23, 24], there is a need to determine how they relate to and influence depression or anxiety in the pandemic context. Thus, understanding which of these factors and what other sociodemographic characteristics or experiences linked to the pandemic are associated with the presence of depressive and/or anxiety symptoms at a later timepoint in the pandemic is necessary because as pandemic conditions change, decision-makers require more data to better distribute mental health and/or addiction (MHA) resources.

Substance use disorders also affect many people as global pre-pandemic prevalence estimates cite heavy alcohol use (18%), daily tobacco use (15%), past-year cannabis use (4%), and past-year use of other substances (<1%) [25]. Similar to mental health concerns, rates of substance use problems during the pandemic have increased [26]. A study from the early months

of the pandemic showed 30% of participants reporting harmful and severe levels of alcohol consumption and almost a quarter of adults reported using drugs, with 38% of this group having severe levels of drug use [27]. However, a systematic review suggested the increases in alcohol consumption during the pandemic may be more modest as some studies showed decreased or varied changes in use, although use of other substances has shown more consistent increases overall [28]. A more updated prevalence of substance use in specific regions is required to understand local challenges.

As the pandemic progresses, rates of MHA concerns and factors that promote or sustain them may vary, so there is a continued need for monitoring. This study's objective was to identify the prevalence of depression, anxiety, and substance use problems in the general population of Ontario, Canada at a time point nearly two years into the pandemic. Furthermore, we sought to determine which demographic characteristics and social determinants of health were independently associated with symptoms of depression only, anxiety only, and both depression and anxiety.

## Method

### Settings and participants

This study used data from the Phase 3 COVID-19 Mental health & Addictions Service impacts & Care needs (MASC) Study, which aims to examine the mental health and substance use effects of the COVID-19 pandemic on adults in Ontario, Canada [29]. Data reported in this study were collected from January to March 2022, two years after a state of emergency was declared in Ontario [30]. The inclusion criteria for the study were: ≥18 years of age, living in the province of Ontario, Canada, and being registered with Delvinia's AskingCanadians respondent panel. The exclusion criterion was: respondent's quota full (based on *a priori* quotas for age, gender, and region). The interlocking quota's purpose was to establish representativeness based on age and gender for Ontario residents with differences under five percent of the mean or five percentage points total, as applicable. Despite using regional quotas, less populated regions (outside of Toronto–Ontario's largest city) were purposely oversampled so that in each area, the known prevalence of depression could be estimated within a 5% margin of error based on sufficient sample size so that appropriate conclusions could be reached [31]. Eligible participants were randomly contacted.

The AskingCanadians panel consists of more than 1 million Canadians. The panel is regularly benchmarked against data from Statistics Canada to ensure representativeness, and more than 500 sociodemographic measures are collected to help ensure panelists meet researchers' needs [32]. Please see study protocol for further details regarding recruitment and sampling [29]. All procedures involving human subjects/patients were approved by the Sunnybrook Health Sciences Centre Research Ethics Board (Ref # 1931), and written informed consent was obtained from all participants.

### Measures

Complete details of the following measures can be found in the study protocol [29]. Data were collected on sociodemographic variables (age, gender, ethnicity, marital status, education level, living situation, and geographic region in Ontario), symptoms of depression and anxiety, substance use, and pandemic-related outcomes (COVID-19 infection fear, impact of pandemic on mental health, social support, SES, and MHA service needs).

**MHA outcomes.** *Depression and anxiety*. The Diagnostic and Statistical Manual of Mental Disorders, Fifth Edition, (DSM-5) Self-Rated Level 1 Cross-Cutting Symptom Measure—Adult is a self-reported measure that assesses mental health symptoms from 13 psychiatric

domains over the past two weeks [33]. However, only the Depression and Anxiety domains were used for this study. There are two and three items, respectively, from the Depression and Anxiety domains, and all items are scored on a 5-point (0–4) scale with higher scores suggesting greater problems in that domain [33]. If any item receives a score ≥2, the participant screens positive in that psychiatric domain and should be further evaluated.

*Substance use.* Participants also completed the Alcohol, Smoking and Substance Involvement Screening Test (ASSIST) Version 3.0, which is an 8-item measure assessing risky or problematic use for 10 substance categories [34, 35]. Aside from tobacco products, alcoholic beverages, cannabis, and opioids, remaining substances were classified under the "other substances" category (e.g., cocaine, amphetamine type stimulants, inhalants, sedatives/sleeping pills, hallucinogens, and other) for this study. Specific substance involvement (SSI) scores 0–3 indicate low risk, 4–26 signifies moderate risk, and ≥27 is high risk. The exception is alcohol, whose scoring cutoffs (from low to high risk) are 0–10, 11–26, and ≥27 [34, 35]. Scores were classified into low and moderate/high risk for this study.

**Pandemic-related outcomes.** *COVID-19 infection fear.* Participants commented on their fear of catching COVID-19 themselves or someone close to them contracting it in the past week. Options included not worried, slightly, moderately, very, and extremely worried. Responses were dichotomized as not worried versus worried.

*Impact of pandemic on mental health.* Participants also rated how the pandemic has affected their mental health. Options included unchanged, a little better/worse, a lot better/worse, very much better/worse. Responses were categorized into worse versus no change/better.

*Social support.* Satisfaction with the availability of natural social supports (e.g., friends, family, romantic partners, community groups, coworkers, pets, etc.) since the onset of the pandemic was rated by participants as neither (dis)satisfied, (dis)satisfied, very (dis)satisfied, or extremely (dis)satisfied. Support was dichotomized into satisfied and neither/dissatisfied.

*SES.* The MacArthur Scale of Subjective Social Status-Adult Version was used to assess SES [36]. Participants completed two 10-point (1–10) slider scales, one requiring retrospective recall for the months just preceding the pandemic and one for during the pandemic, asking them to compare their own income, education, and jobs to those around them. A score of 1 was worst off and 10 was best off. The change in SES was calculated as the difference between the pandemic and pre-pandemic period, and results were classified as worse off and no change/better off.

*MHA service needs.* Thoughts of using MHA services since the pandemic was declared were reported by participants. Services encompassed but were not limited to sessions with healthcare providers, crisis supports, residential treatment, support groups, and MHA programs. Responses were grouped into having needs unmet (thought about accessing services but did not) and needs met (not needing services/needing and accessing services).

**Statistical analysis.** The whole sample was divided into four groups, namely subjects with depression only, anxiety only, both depression and anxiety, and neither. For categorical variables, 2x4 contingency tables were created, and chi-squared ($\chi^2$) tests were used to test for significance. For statistically significant models, 2x2 contingency tables and $\chi^2$ tests were then performed to identify between-group differences. For continuous variables, one-way analysis of variance (ANOVA) or multivariate analysis of variance (MANOVA) were used. If results were statistically significant, post hoc analyses using Fisher's least significant difference test were performed to identify between-group differences.

A multinomial logistic regression model was then constructed to determine which variables were independently associated with screening positive for symptoms of depression only, anxiety only, and both depression and anxiety (reference group was those with neither depression nor anxiety). The included variables were sociodemographic characteristics (i.e., age groups,

gender, ethnicity, marital status, education level, and living situation), ASSIST substance use risk levels (low versus moderate/high) [34, 35] for tobacco, alcohol, and cannabis, and all pandemic-related outcomes. Independent variables were simultaneously added into the model. Statistical analyses were performed using SPSS Statistics 28 (IBM Corporation). Statistically significant results were defined as $p < .05$ (two-tailed).

## Results

In total, 5000 participants completed the survey. The average age was 47.9 (SD 16.3) years old, with an equal number of male and female participants. Most were White (69.8%), married/in a common-law relationship (61.4%), had some postsecondary education or more (87.0%), and living with others (78.0%). For participant characteristics, there were statistically significant differences between groups for age, gender, ethnicity, and marital status but no consistent pattern was found. Table 1 presents full descriptive characteristics.

According to census data from Statistics Canada for 2022, the sample was representative based on age and sex. The mean age of participants was 47.9 years (within 5% of the mean age of those ≥18 years old living in Ontario, which is 48.3 years), and our sample was comprised

**Table 1. Baseline characteristics of participants[1].**

| | Total (*n* = 5000) | Depression Only (*n* = 413) | Anxiety Only (*n* = 534) | Both Depression and Anxiety (*n* = 1786) | Neither (*n* = 2267) | df | F | Sig. |
|---|---|---|---|---|---|---|---|---|
| **Mean age (SD)** | 47.9 (16.3) | 48.8 (15.8)[a] | 44.4 (15.8)[b] | 42.5 (14.8)[c] | 52.9 (16.0)[d] | 3 | 159.235 | <0.001* |
| **Gender[2]** | | | | | | | Value | |
| Male | 2465 (50.0) | 225 (54.9)[a] | 214 (40.6)[b] | 785 (45.2)[b] | 1241 (55.0)[a] | 3 | 60.804 | <0.001* |
| Female | 2465 (50.0) | 185 (45.1) | 313 (59.4) | 951 (54.8) | 1016 (45.0) | | | |
| **Ethnicity** | | | | | | | | |
| White | 3491 (69.8) | 279 (67.6)[a, d] | 358 (67.0)[a, b] | 1217 (68.1)[a, b] | 1637 (72.2)[c, d] | 3 | 11.497 | 0.01* |
| Non-White | 1509 (30.2) | 134 (32.4) | 176 (33.0) | 569 (31.9) | 630 (27.8) | | | |
| **Marital status** | | | | | | | | |
| Married/Common-law | 3072 (61.4) | 250 (60.5)[a] | 319 (59.7)[a] | 991 (55.5)[a] | 1512 (66.7)[b] | 3 | 53.947 | <0.001* |
| Divorced/Widowed/ Single | 1928 (38.6) | 163 (39.5) | 215 (40.3) | 795 (44.5) | 755 (33.3) | | | |
| **Level of education** | | | | | | | | |
| High school or less | 649 (13.0) | 55 (13.3) | 63 (11.8) | 223 (12.5) | 308 (13.6) | 3 | 1.826 | 0.61 |
| Some postsecondary or more | 4351 (87.0) | 358 (86.7) | 471 (88.2) | 1563 (87.5) | 1959 (86.4) | | | |
| **Living situation** | | | | | | | | |
| Living alone | 1101 (22.0) | 95 (23.0) | 109 (20.4) | 416 (23.3) | 481 (21.2) | 3 | 3.570 | 0.31 |
| Living with others | 3899 (78.0) | 318 (77.0) | 425 (79.6) | 1370 (76.7) | 1786 (78.8) | | | |
| **Geographical region** | | | | | | | | |
| Toronto | 750 (15.0) | 64 (15.5) | 75 (14.0) | 261 (14.6) | 350 (15.4) | 12 | 15.284 | 0.23 |
| Southwestern Ontario | 1250 (25.0) | 101 (24.5) | 133 (24.9) | 475 (26.6) | 541 (23.9) | | | |
| Eastern Ontario | 1000 (20.0) | 87 (21.1) | 113 (21.2) | 380 (21.3) | 420 (18.5) | | | |
| Central Ontario | 1500 (30.0) | 126 (30.5) | 160 (30.0) | 495 (27.7) | 719 (31.7) | | | |
| Northern Ontario | 500 (10.0) | 35 (8.5) | 53 (9.9) | 175 (9.8) | 237 (10.5) | | | |

[1] reported as *n* (% within column) unless otherwise stated

[2] non-binary-identifying individuals (*n* = 70) were excluded due to small sample size

[a, b, c, d] within each row, groups not sharing letters differ significantly at $p < .05$

* $p < .05$

of 49.3% male and 49.3% female participants (both within 5% of the sex proportions of Ontario, which is 49.1% male and 50.9% female) [37].

## Prevalence of MHA concerns and pandemic-related outcomes (Table 2)

The number of participants screening positive for depression only (8.3%), anxiety only (10.7%), both depression and anxiety (35.7%), and neither (45.3%) are presented in Table 2. Moderate to high risk levels of substance use were found in 19.7% of participants for tobacco, 16.8% for alcohol, 17.0% for cannabis, 3.4% for opioids, and 4.7% for other substances. Overall, 66.1% worried about themselves or someone close contracting COVID-19, 55.0% believed the pandemic negatively impacted their mental health, 48.7% were satisfied with their social

**Table 2. Depression, anxiety, substance use, and COVID-19-related outcomes.**

| | Total (n = 5000) | Depression Only (n = 413) | Anxiety Only (n = 534) | Both Depression and Anxiety (n = 1786) | Neither (n = 2267) | df | F | Sig. |
|---|---|---|---|---|---|---|---|---|
| **DSM-5 Self-Rated Level 1 Cross-Cutting Symptom Measure–Adult, mean item score (SD)[1]** | | | | | | | | |
| Depression domain | 1.2 (1.1) | 1.7 (0.6)[a] | 0.7 (0.4)[b] | 2.4 (0.8)[c] | 0.3 (0.4)[d] | 5 | 2553.494 | <0.001* |
| Anxiety domain | 1.1 (1.0) | 0.5 (0.4)[a] | 1.4 (0.5)[b] | 2.1 (0.8)[c] | 0.3 (0.3)[d] | 5 | 2176.104 | <0.001* |
| **ASSIST SSI scores, mean (SD)[1]** | | | | | | | | |
| Tobacco | 3.1 (6.6) | 3.0 (6.2)[a] | 2.6 (5.9)[a, d] | 4.4 (7.9)[b] | 2.3 (5.7)[c, d] | 5 | 25.349 | <0.001* |
| Alcohol | 5.6 (6.4) | 5.4 (5.6)[a] | 5.7 (5.8)[a] | 7.3 (8.0)[b] | 4.4 (4.6)[c] | 5 | 58.822 | <0.001* |
| Cannabis[2] | 2.1 (5.3) | 1.6 (4.2)[a] | 2.0 (4.9)[a] | 3.7 (7.1)[b] | 1.0 (3.3)[c] | 5 | 94.743 | <0.001* |
| Opioids | 0.5 (2.8) | 0.3 (2.0)[a] | 0.3 (1.9)[a] | 0.9 (4.1)[b] | 0.2 (1.7)[a] | 5 | 17.501 | <0.001* |
| Other substances[3] | 0.6 (3.3) | 0.4 (2.2)[a] | 0.4 (2.1)[a] | 1.3 (4.8)[b] | 0.2 (1.7)[a] | 5 | 33.720 | <0.001* |
| **ASSIST risk level (moderate/high),[4] n (% yes within column)** | | | | | | | Value | |
| Tobacco | 987 (19.7) | 80 (19.4)[a] | 98 (18.4)[a, d] | 470 (26.3)[b] | 339 (15.0)[c, d] | 3 | 82.210 | <0.001* |
| Alcohol | 839 (16.8) | 70 (16.9)[a] | 89 (16.7)[a] | 456 (25.5)[b] | 224 (9.9)[c] | 3 | 175.249 | <0.001* |
| Cannabis[2] | 849 (17.0) | 60 (14.5)[a] | 89 (16.7)[a] | 498 (27.9)[b] | 202 (8.9)[c] | 3 | 257.143 | <0.001* |
| Opioids | 169 (3.4) | 12 (2.9)[a] | 13 (2.4)[a, d] | 111 (6.2)[b] | 33 (1.5)[c, d] | 3 | 71.407 | <0.001* |
| Other substances[3] | 234 (4.7) | 12 (2.9)[a] | 20 (3.7)[a] | 168 (9.4)[b] | 34 (1.5)[c] | 3 | 144.797 | <0.001* |
| **COVID-19 infection fear, n (% worried within column)** | 3306 (66.1) | 264 (63.9)[a] | 381 (71.3)[b, d] | 1295 (72.5)[c, d] | 1366 (60.3)[a] | 3 | 74.745 | <0.001* |
| **Impact of pandemic on mental health, n (% worse within column)** | 2748 (55.0) | 274 (66.3)[a] | 333 (62.4)[a] | 1454 (81.4)[b] | 687 (30.3)[c] | 3 | 1094.956 | <0.001* |
| **Social support, n (% satisfied within column)** | 2436 (48.7) | 186 (45.0)[a] | 281 (52.6)[b] | 626 (35.1)[c] | 1343 (59.2)[d] | 3 | 239.522 | <0.001* |
| **Socioeconomic status, n (% worse off within column)** | 1416 (28.3) | 118 (28.6)[a] | 158 (29.6)[a] | 693 (38.8)[b] | 447 (19.7)[c] | 3 | 179.739 | <0.001* |
| **Unmet MHA service needs, n (% yes within column)** | 1243 (24.9) | 83 (20.1)[a] | 157 (29.4)[b] | 832 (46.6)[c] | 171 (7.5)[d] | 3 | 826.089 | <0.001* |

Abbreviations: ASSIST, Alcohol, Smoking and Substance Involvement Screening Test; DSM-5, Diagnostic and Statistical Manual of Mental Disorders, Fifth Edition; MHA, mental health and/or addiction; SSI, specific substance involvement

* $p < .05$

[a, b, c, d] Within each row, groups not sharing letters differ significantly at $p < .05$

[1] n = 4930 because non-binary individuals were not included in the multivariate analysis of variance (MANOVA)

[2] Not prescribed by a physician

[3] E.g., cocaine, amphetamines, inhalants, sedatives/sleeping pills, hallucinogens, and other drugs (not prescribed or used as prescribed)

[4] SSI scores ≥4 meet the cutoff for moderate/high risk use with the exception of alcohol (≥10)

supports during the pandemic, 28.3% believed they were worse off socioeconomically since the pandemic began, and 24.9% had unmet MHA service needs during the pandemic.

For both the Depression and Anxiety domain scores as well as the ASSIST SSI scores, the models were statistically significant even with age and gender as covariates in the MANOVA analysis. Each of the four groups' scores were statistically significantly different from each other for the Depression and Anxiety domains. For both domains, those with both depression and anxiety scored the highest, while those with neither scored the lowest. For ASSIST SSI scores, ASSIST risk levels, and pandemic-related outcomes, there were statistically significant differences in scores between groups, but no consistent pattern was found. However, in general, those with both depression and anxiety had the worst outcomes, followed by those with depression or anxiety only, and finally those with neither.

## Factors associated with positive screening for symptoms of depression only, anxiety only, and both depression and anxiety (Table 3)

In total, 4930 participants were included in the multinomial logistic regression model. Those with neither depression nor anxiety were used as the reference group. Data from non-binary-identifying participants ($n$ = 70) were not included due to small sample size. The regression model for factors associated with positive screening for symptoms of depression only, anxiety only, and both depression and anxiety was significant ($\chi^2(48)$ = 2125.590, $p < .001$), explaining 38.9% of the variance (using Nagelkerke $R^2$) (see Table 3).

Compared to the neither depression nor anxiety group, the depression only group was more likely to be BIPOC (Black, Indigenous, and People of Color) (OR = 1.35, 95% CI = 1.05–1.74, $p$ = .02), have moderate/high risk levels of alcohol use (OR = 1.41, 95% CI = 1.03–1.93, $p$ = .03), have negative change in mental health since the pandemic started (OR = 4.07, 95% CI = 3.22–5.14, $p < .001$), and have unmet MHA service needs (OR = 2.19, 95% CI = 1.61–2.97, $p < .001$). Satisfaction with social support decreased the likelihood of being in the depression only group (OR = 0.52, 95% CI = 0.44–0.61, $p$ = .002) as compared to the neither depression nor anxiety group. No other independent variables were significant.

The anxiety only group was more likely to be in the 18–34 (OR = 2.87, 95% CI = 2.09–3.94, $p < .001$) and 35–48 age groups (OR = 1.88, 95% CI = 1.39–2.56, $p < .001$), be female (OR = 1.64, 95% CI = 1.34–2.02, $p < .001$), have moderate/high risk levels of alcohol use (OR = 1.53, 95% CI = 1.15–2.05, $p$ = .004), have COVID-19 infection fear (OR = 1.45, 95% CI = 1.16–1.81, $p < .001$), have negative change in mental health since the pandemic started (OR = 3.12, 95% CI = 2.52–3.85, $p < .001$), and have unmet MHA service needs (OR = 3.01, 95% CI = 2.32–3.91, $p < .001$) compared to the neither depression nor anxiety group. No other independent variables were significant.

Compared to the group with neither depression nor anxiety, the group screening positive for both depression and anxiety was more likely to be in the 18–34 (OR = 3.06, 95% CI = 2.38–3.94, $p < .001$), 35–48 (OR = 2.10, 95% CI = 1.65–2.67, $p < .001$), and 49–61 (OR = 1.69, 95% CI = 1.34–2.13, $p < .001$) age groups; be female (OR = 1.27, 95% CI = 1.09–1.49, $p$ = .003); have moderate/high risk levels of tobacco (OR = 1.40, 95% CI = 1.13–1.72, $p$ = .002), alcohol (OR = 1.98, 95% CI = 1.59–2.47, $p < .001$), and cannabis (OR = 1.80, 95% CI = 1.43–2.27, $p < .001$) use; have COVID-19 infection fear (OR = 1.53, 95% CI = 1.29–1.82, $p < .001$); have negative change in mental health since the pandemic started (OR = 6.91, 95% CI = 5.83–8.20, $p < .001$); have negative change in SES since the pandemic started (OR = 1.44, 95% CI = 1.21–1.71, $p < .001$); and have unmet MHA service needs (OR = 5.32, 95% CI = 4.33–6.55, $p < .001$). Positive screening for symptoms of both depression and anxiety was less likely to occur for those more satisfied with social supports during the pandemic (OR = 0.71, 95% CI = 0.57–0.88,

**Table 3. Associations between independent variables and positive screening in mental health symptoms (with respect to neither depression nor anxiety group)** ($n$ = 4930).

| Variables | Depression only | | | Anxiety only | | | Both Depression and Anxiety | | |
|---|---|---|---|---|---|---|---|---|---|
| | OR | 95% CI | | OR | 95% CI | | OR | 95% CI | |
| **Demographics** | | | | | | | | | |
| Age group (years) | | | | | | | | | |
| 18–34 | 1.35 | 0.96 | 1.92 | 2.87* | 2.09 | 3.94 | 3.06* | 2.38 | 3.94 |
| 35–48 | 1.16 | 0.83 | 1.60 | 1.88* | 1.39 | 2.56 | 2.10* | 1.65 | 2.67 |
| 49–61 | 1.25 | 0.92 | 1.69 | 1.26 | 0.92 | 1.72 | 1.69* | 1.34 | 2.13 |
| ≥62 | Reference | | | Reference | | | Reference | | |
| Unmarried/widowed | 1.19 | 0.88 | 1.59 | 1.09 | 0.84 | 1.41 | 1.22 | 0.99 | 1.50 |
| Married/Common-law | Reference | | | Reference | | | Reference | | |
| Female | 0.92 | 0.74 | 1.15 | 1.64* | 1.34 | 2.02 | 1.27* | 1.09 | 1.49 |
| Male | Reference | | | Reference | | | Reference | | |
| Non-White | 1.35* | 1.05 | 1.74 | 1.09 | 0.87 | 1.37 | 1.15 | 0.96 | 1.38 |
| White | Reference | | | Reference | | | Reference | | |
| Some postsecondary education or more | 0.92 | 0.66 | 1.28 | 1.01 | 0.74 | 1.38 | 0.96 | 0.76 | 1.23 |
| High school education or less | Reference | | | Reference | | | Reference | | |
| Living alone | 1.07 | 0.77 | 1.48 | 1.01 | 0.75 | 1.37 | 1.15 | 0.91 | 1.45 |
| Living with others | Reference | | | Reference | | | Reference | | |
| **Substances** | | | | | | | | | |
| Moderate/high tobacco risk | 1.17 | 0.87 | 1.57 | 1.12 | 0.84 | 1.47 | 1.40* | 1.13 | 1.72 |
| Low tobacco risk | Reference | | | Reference | | | Reference | | |
| Moderate/high alcohol risk | 1.41* | 1.03 | 1.93 | 1.53* | 1.15 | 2.05 | 1.98* | 1.59 | 2.47 |
| Low alcohol risk | Reference | | | Reference | | | Reference | | |
| Moderate/high cannabis risk | 1.17 | 0.83 | 1.65 | 1.26 | 0.93 | 1.71 | 1.80* | 1.43 | 2.27 |
| Low cannabis risk | Reference | | | Reference | | | Reference | | |
| **Pandemic-related factors** | | | | | | | | | |
| Fear of COVID-19 infection | 1.02 | 0.81 | 1.28 | 1.45* | 1.16 | 1.81 | 1.53* | 1.29 | 1.82 |
| No fear of COVID-19 infection | Reference | | | Reference | | | Reference | | |
| Negative change in mental health since start of pandemic | 4.07* | 3.22 | 5.14 | 3.12* | 2.52 | 3.85 | 6.91* | 5.83 | 8.20 |
| Positive/no change in mental health since start of pandemic | Reference | | | Reference | | | Reference | | |
| Satisfied with social supports | 0.52* | 0.44 | 0.61 | 0.95 | 0.77 | 1.16 | 0.71* | 0.57 | 0.88 |
| Dissatisfied/ambivalent about social supports | Reference | | | Reference | | | Reference | | |
| Negative change in SES since start of pandemic | 1.16 | 0.90 | 1.49 | 1.18 | 0.94 | 1.49 | 1.44* | 1.21 | 1.71 |
| Positive/no change in SES since start of pandemic | Reference | | | Reference | | | Reference | | |
| Unmet MHA service needs | 2.19* | 1.61 | 2.97 | 3.01* | 2.32 | 3.91 | 5.32* | 4.33 | 6.55 |
| Met MHA service needs | Reference | | | Reference | | | Reference | | |
| **Full model** | $\chi^2(48)$ = 2125.590, $p < .001$, Nagelkerke $R^2$ = .39 | | | | | | | | |

Abbreviations: CI, confidence interval; MHA, mental health and/or addiction; OR, odds ratio; SES, socioeconomic status

* $p < .05$

$p < .001$) as compared to the neither depression nor anxiety group. No other independent variables were significant.

## Discussion

This study suggests that the number of people screening positive for symptoms of depression and anxiety, and experiencing substance use problems remained concerning two years into

the pandemic. Results from the multinomial logistic regression indicate that for those with depression only, being BIPOC, having moderate/high risk levels of alcohol use, negative change in mental health since the pandemic started, and unmet MHA service needs were predictors, while for those with anxiety only, younger age, female gender, moderate/high risk levels of alcohol use, and all pandemic-related factors (except for negative change in SES and satisfaction with social supports) were predictors. Meanwhile, younger age, female gender, moderate/high risk levels of substance use, and negative pandemic-related variable outcomes were significantly and independently associated with positive screening for symptoms of both depression and anxiety. Increased satisfaction with social supports reduced the likelihood of being in the depression only and both depression and anxiety groups, but not the anxiety only group. Remaining demographic characteristics were not significantly independently associated with the depression only, anxiety only, or both depression and anxiety groups.

The prevalence rate of screening positive for depressive and anxiety symptoms regardless of comorbidity in this study was higher than many findings reported early in the pandemic, suggesting mental health concerns remained a substantial problem for many people [19–21]. While there is evidence that mental health symptoms may have subsided in the first half year of the pandemic after an initial spike [9, 19], more long-term studies have found symptom increases [7, 8]. Prevalence rates in this study may be greater than previous reports because the data collection period (January to March 2022) coincided with the 5th wave of COVID-19 in Canada, which saw the greatest weekly case counts of the entire pandemic to that point [38]. This circumstance may have contributed to heightened rates of symptoms of depression and anxiety, such as from fear of COVID-19 infection to the re-introduction of restrictive public health measures. Thus, this study may have captured a temporary increase in rates of screening positive for depression and anxiety. Another potential explanation for the higher prevalence rates in the present study compared to studies from earlier in the pandemic may be the use of the DSM-5 measurement tool in the current study. A prior study that administered the same measure in a general population sample in Brazil during the pandemic found very high prevalence rates of screening positive for symptoms of depression (68%) and anxiety (82%) [39]. Nevertheless, this tool was designed to aid clinicians in identifying people at risk for various adverse mental health outcomes, so the present study may have identified a pressing need regardless. Concurrently, increased prevalence rates in the present study may be due to increased awareness and visibility surrounding mental health issues during the pandemic.

Risky substance use, suggesting harmful substance use and/or dependence, was also identified. Moderate/high risk levels of alcohol and tobacco use in our study were comparable to pre-pandemic estimates of heavy alcohol and tobacco use identified by Peacock et al. [25], although moderate/high risk levels of cannabis, opioids, and other substance use were higher in our study. Elevated rates of risky cannabis use in this study may have resulted from this substance being legal to purchase and use recreationally in Canada (as of fall 2018). With regards to opioids, poisonings and substance use disorders have steadily increased in recent years in Canada [40]. Alternatively, it may be that greater risk levels with substance use are related to experiencing more social isolation that resulted from lockdowns [41]. Substance use problems may have increased or remained at concerning levels because in Ontario, various severity levels of public health measures were implemented due to emerging SARS-CoV-2 variants and changing COVID-19 case counts, with certain regions experiencing some of the longest-standing restrictions in the world at the time [42]. Moreover, with millions of healthcare services being backlogged and harm reduction/treatment services being disrupted by the pandemic [43], an accumulation of patients from earlier in the pandemic may have delayed the ability to access addiction services and exacerbated increasing trends of substance use problems. That said, although this study identified rates of risky substance use that were higher than pre-

pandemic levels, these rates were lower than reports from the early stages of the pandemic [26, 27], perhaps because people may have been using substances to cope with the uncertainty initially surrounding the pandemic [44]. However, as more information was available on the pandemic and public health measures were lifted, other forms of enduring the adversities of the pandemic became more available, such as social gatherings, so the way people interacted with substances may have changed.

The multinomial logistic regression results in this study did not support previous research that identified associations between demographic characteristics (including age, gender, ethnicity, living situation, and marital status) and depression or anxiety [19–22]. With the exception of younger age and female gender, which was associated with higher likelihood of being in the anxiety only and both depression and anxiety groups, and ethnicity, which was associated with the depression only group, other demographic characteristics (i.e., marital status, education level, and living situation) were not statistically significantly associated. Perhaps these characteristics were important initially during the pandemic in identifying those at risk for depressive and/or anxiety symptoms, but the impact of these factors may have changed over time. It may be that the pandemic disproportionately impacted people with certain demographic characteristics at the onset of the pandemic, compared with during the so-called 5th wave. For instance, COVID-19 infection risk was increased among racialized and/or marginalized groups [45], and isolation/loneliness may have been more pronounced for single individuals and those living alone during social distancing and lockdown measures [46, 47]. However, as the pandemic progressed, vaccines became more readily available and restrictions were lifted [48, 49], so these disparities from the early part of the pandemic may have been reduced. Alternatively, many pandemic-related variables, which prior studies have not all examined, were added into the regression model. Thus, statistical power may have shifted from demographic characteristics to these additional variables. Female gender may have been associated with being in the anxiety only and both anxiety and depression groups because women were disproportionately impacted by such things as job losses during the pandemic, and women with children were more likely to have dropped out of the workforce [4]. Female caregivers are known to have taken on more caregiving and household responsibilities during the pandemic [4]; thus, stress from worrying about their children's schooling and health or the health of elderly family members may have contributed to increased rates of anxiety symptoms [50]. Moderate/high risk levels of all substance use were also associated with the group that had both depression and anxiety, while only moderate/high risk levels of alcohol use were also associated with the depression only and anxiety only groups. Considering mental health issues and substance use problems are highly comorbid, it is unsurprising to see this association in our findings [51]. In addition, pandemic-related variables were all independently associated with the group that had both depression and anxiety but not the groups with depression or anxiety alone. Of note, those who were satisfied with their social supports were less likely to be in the depression only and both depression and anxiety groups, while those with unmet MHA needs were much more likely to be in the depression only, anxiety only, or both depression and anxiety groups. More perceived social supports has been found to be associated with greater psychological resilience [52], which may explain how social support can act as a protective factor. Meanwhile, unmet MHA service needs indicate people are not receiving the treatment and care they require, so existing mental health concerns may worsen. Understanding the specific reasons people could not/did not access services should be explored in future studies and will be helpful in addressing those barriers. Lastly, it appears that for the most part, the group with both depression and anxiety was more severely impacted both in terms of the number of outcomes and the strength of association. This finding is in keeping with prior studies which have found that those with both depression and anxiety experience worse outcomes

(e.g., greater symptom severity, more disability, poorer work performance, etc.) compared to those with either depression or anxiety only or neither [53–55]. Perhaps these participants were experiencing a greater number of symptoms due to an additive effect from depression and anxiety.

This study has several strengths and some limitations. Participants were recruited based on predetermined quotas for age, gender, and geographic region to ensure a representative sample of Ontario residents, improving the generalizability of the findings. Additionally, the large sample size allowed improved power over previous studies in the field. Many pandemic-related variables were simultaneously captured in the survey, which allowed for an investigation into various factors that predispose people to increased likelihood for screening positive for symptoms of depression and/or anxiety. However, differences in rates of depression, anxiety, and substance use compared to pre-pandemic/early pandemic estimates should be interpreted with caution. As these previous reports relied on different methodologies and sampled in other geographic regions, discrepancies in results may be a function of different assessment tools and/or populations. Furthermore, populations anticipated to have lower response rates (such as in Northern Ontario which is more rural) were oversampled so that the prevalence of mental health symptoms could be estimated within a 5% margin of error. However, this over-sampling may also reduce the representativeness of our results, and there may also have been self-selection bias, so caution should be exercised when interpreting these findings. Moreover, the reliance on retrospective recall for pre-pandemic variables may have introduced bias. Additionally, only individual and not contextual variables were explored, which may lead to an individualistic fallacy whereby an assumption is made that individual-level outcomes can only be explained by individual-level variables [56]. Lastly, the cross-sectional design of the study did not allow for a longitudinal investigation, meaning this study can provide insight into associations only and cannot comment on or assign causality. As prevalence estimates of MHA concerns and the factors associated with positive screening for depression and/or anxiety are likely to change over time, investigations should be repeated as the pandemic progresses.

MHA concerns continued to be a burden two years into the pandemic. Risky alcohol use and certain pandemic-related factors (e.g., negative change in mental health, unmet MHA service needs) were associated with positive screening for symptoms of depression only, anxiety only, and both depression and anxiety more so than demographic characteristics. Front-line providers working with patients and their families should remain aware of the potential for elevated MHA concerns, particularly among those who have unmet MHA service needs.

## Author Contributions

**Conceptualization:** Anthony Levitt, Sugy Kodeeswaran, Roula Markoulakis.

**Data curation:** Maida Khalid, Roula Markoulakis.

**Formal analysis:** Oswin Chang, Maida Khalid, Roula Markoulakis.

**Funding acquisition:** Anthony Levitt, Sugy Kodeeswaran, Roula Markoulakis.

**Investigation:** Anthony Levitt, Sugy Kodeeswaran, Roula Markoulakis.

**Methodology:** Anthony Levitt, Sugy Kodeeswaran, Roula Markoulakis.

**Project administration:** Anthony Levitt, Roula Markoulakis.

**Resources:** Anthony Levitt, Sugy Kodeeswaran.

**Supervision:** Anthony Levitt, Roula Markoulakis.

**Validation:** Anthony Levitt, Maida Khalid, Sugy Kodeeswaran, Roula Markoulakis.

**Writing – original draft:** Oswin Chang.

**Writing – review & editing:** Oswin Chang, Anthony Levitt, Maida Khalid, Sugy Kodeeswaran, Roula Markoulakis.

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
