## [Decision Letter · Decision Letter 0]

6 Nov 2023

PONE-D-23-30089The prevalence and predictors of mental health and addiction concerns during the COVID-19 pandemic in Ontario, Canada: A cross-sectional studyPLOS ONE

Dear Dr. Markoulakis,

Thank you for submitting your manuscript to PLOS ONE. After careful consideration, we feel that it has merit but does not fully meet PLOS ONE’s publication criteria as it currently stands. Therefore, we invite you to submit a revised version of the manuscript that addresses the points raised during the review process.

We look forward to receiving your revised manuscript.

Kind regards,

Guillermo Salinas-Escudero, PhD. MsC.

Academic Editor

PLOS ONE

Journal Requirements:

This work was supported by the Canadian Institutes of Health Research (AL, SK, RM; grant number UIP-178831) and the Hurvitz Brain Sciences Summer Student Research Program (OC) through the Sunnybrook Research Institute. The funders had no role in study design, data collection and analysis, decision to publish, or preparation of the manuscript. 

Reviewers' comments:

Reviewer's Responses to Questions

**Comments to the Author**

1. Is the manuscript technically sound, and do the data support the conclusions?

Reviewer #1: No

Reviewer #2: Partly

2. Has the statistical analysis been performed appropriately and rigorously? 

Reviewer #1: No

Reviewer #2: Yes

3. Have the authors made all data underlying the findings in their manuscript fully available?

Reviewer #1: Yes

Reviewer #2: Yes

4. Is the manuscript presented in an intelligible fashion and written in standard English?

Reviewer #1: Yes

Reviewer #2: Yes

5. Review Comments to the Author

Reviewer #1: It is necessary adequately define the objective of the study, what specific result you want to present and redo the analysis in that sense. What does this article contribute to?

Introduction:

It covers many topics without delving into any of them, prevalence, predictors, disparities…. It is not clear what he calls “mental health concerns”.

Methodology

It is not described, everything refers to another document, it is not possible for the reader to understand the basic source of information or methodology with what is provided in the section.

1) It seems to me that there was no reflection on the results to be presented, there is no robust conceptual or theoretical presentation. It is not defined what they call MHA or pandemic related outcomes.

2) how they model the socioeconomic condition, to adequately isolate the symptoms from what is attributable to the experience with the COVID pandemic.

Omissions with potentially serious biases in the prevalence measurement:

• They do not include if people already had a diagnosis and/or received treatment for the mental health condition, then, prevalence of what the survey measures? people with symptoms and not treated, or treated inappropriately?

• It is not described what they include as variables associated with covid, the person or close family member had COVID, was they hospitalized or even died from this cause?, whether they have already received the vaccine or not.

• The presence of symptoms of depression and anxiety represents 35.7% of the study population, and authors omits to analyze this population!!

- Using the word predictors is wrong as it comes from a cross-sectional survey

2) Why ruin all the work of collecting a large sample by dividing the statistical analysis of the sample into two parts. Why a separate regression model for anxiety and another for depression, what justification does this strategy have, and again why does it not model the depression-anxiety comorbidity?

3) The regression models seem poorly worked to me, examples:

• What was the point of seeing in Table 1 that there are differentiated patterns by age, if the multivariate model includes the age variable as a linear continuous, without any type of adjustment.

• What the variables Fear of covid, unmet MHA service needs mean.

Results

Table 1: It is necessary to reflect on what is presented

- How useful are the columns (not depression and not anxiety)?

- Why population with anxiety and depression symptoms are no presented?

- I think these columns should be contrast: Total population, Pop with symptoms of depression, Pop with symptoms of anxiety, Pop with symptoms of both, and Pop without symptoms

Table 2: I do not understand this table,

- Why not present levels or severity of symptoms instead of presenting averages.

- How useful is it to compare the average response of the non-depressed population with depression, if the measuring instrument to classify in column as in row was the same

Table 3 and 4

It seems absurd to me to divide the analysis into two models, the study variable is mental health at the population level. It is necessary to reflect on how to properly model

Reviewer #2: The study reports findings on mental health and addiction concerns on adults in Ontario, Canada. This issue is important globally due to its possible increase during and after the COVID-19 pandemic.

The study design is cross-sectional and the attribution of causality with the pandemic has important limitations inherent to the data. Therefore, it is important that the authors point out more clearly that the increase in the occurrence of mental health and addiction concerns may be due to the pandemic or due to greater visibility of mental health issues (better and greater diagnostic coverage).

From a methodological point of view, the statistical analysis did not take advantage of the possibility of exploring contextual effects that seem to have had an impact on the findings. See table 1 for the differences between Southwestern Ontario, Eastern Ontario and Central Ontario. It would be interesting to know the results of a multilevel analysis, since the findings on the protective association of social support are suggestive of differential social processes between the regions.

6. PLOS authors have the option to publish the peer review history of their article (what does this mean?). If published, this will include your full peer review and any attached files.

Reviewer #1: No

Reviewer #2: No

---

## [Author Response · Author response to Decision Letter 0]

8 Jan 2024

Please see attached "Response to Reviewers" file.

---

## [Decision Letter · Decision Letter 1]

30 Jan 2024

PONE-D-23-30089R1The prevalence of mental health and addiction concerns and factors associated with depression and anxiety during the COVID-19 pandemic in Ontario, Canada: A cross-sectional studyPLOS ONE

Dear Dr. Markoulakis,

Thank you for submitting your manuscript to PLOS ONE. After careful consideration, we feel that it has merit but does not fully meet PLOS ONE’s publication criteria as it currently stands. Therefore, we invite you to submit a revised version of the manuscript that addresses the points raised during the review process.

I agree with the opinion of both reviewers. The authors should have delved into the answers to the questions made by the reviewers. Therefore, the next version needs to include the required changes and adequately support the content of the work; otherwise, I suggest rejecting the work for publication.

We look forward to receiving your revised manuscript.

Kind regards,

Guillermo Salinas-Escudero, PhD. MsC.

Academic Editor

PLOS ONE

Additional Editor Comments:

I agree with the opinion of both reviewers. The authors should have delved into the answers to the questions made by the reviewers. Therefore, the next version needs to include the required changes and adequately support the content of the work; otherwise, I suggest rejecting the work for publication.

Reviewers' comments:

Reviewer's Responses to Questions

**Comments to the Author**

1. If the authors have adequately addressed your comments raised in a previous round of review and you feel that this manuscript is now acceptable for publication, you may indicate that here to bypass the “Comments to the Author” section, enter your conflict of interest statement in the “Confidential to Editor” section, and submit your "Accept" recommendation.

Reviewer #1: All comments have been addressed

Reviewer #2: All comments have been addressed

2. Is the manuscript technically sound, and do the data support the conclusions?

Reviewer #1: Partly

Reviewer #2: Partly

3. Has the statistical analysis been performed appropriately and rigorously? 

Reviewer #1: No

Reviewer #2: Yes

4. Have the authors made all data underlying the findings in their manuscript fully available?

Reviewer #1: Yes

Reviewer #2: Yes

5. Is the manuscript presented in an intelligible fashion and written in standard English?

Reviewer #1: Yes

Reviewer #2: Yes

6. Review Comments to the Author

Reviewer #1: Although the authors responded to each observation, substantial aspects were left unaddressed.

The study could provide information on the prevalence of depression and anxiety symptoms, 2 years after the appearance of COVID-19 and contact restrictions. However, with the information added by the authors, in this second review I doubt whether the study is representative of the target population, since they mentioned that it is a non-random sample.

The methodological description of population selection remains incomplete. Crucial aspects to understand how the sample was selected were omitted. In the response they showed that the sampling was a non-random sampling by quotas, this aspect totally changes the external validity of the study, and it would be necessary for the authors to show simple evidence of the representativeness of the sample, in the same document, and not just reference other documents.

Another aspect about the selection is that they report that they were participants registered in “Delvinia’s AskingCanadians responder”, without explaining in the slightest or making any reference to it.

Another important aspect is that, given the observation that the analysis "independently" analyzes the population with two of the health conditions, the authors only respond "because it is of interest to them." However, they do not support this assumption of independence on background, nor do they do any statistical treatment to show what they are talking about. I find it questionable that they show blindness to a large proportion of the population who report both health conditions.

Regarding the statistical analysis, the observation was made that there are differentiated effects by age groups, but in the regression analysis the authors only included the continuous variable, and it is not mentioned that non-linearities were evaluated. And the authors only responded that there were already many categorical variables.

Due to the above, I think that the article only provides information on prevalence, since the association analysis is still very incipient.

Reviewer #2: The new version included many recommendations from reviewers and is better than the previous one. However, it has an inconsistency between the introduction and analysis of the data. In the introduction it is indicated that the context can be a determinant of outcomes, but it is not explored. Only individual data is analyzed. The authors' explanation for not doing multilevel analysis is insufficient. It is suggested that the authors point out as a limitation that only individual variables and not contextual variables were explored, which can lead to a psychological fallacy (see Diez-Roux AV. Bringing context back into epidemiology: variables and fallacies in multilevel analysis. Am J Public Health 1988;88(2):216-22.

7. PLOS authors have the option to publish the peer review history of their article (what does this mean?). If published, this will include your full peer review and any attached files.

Reviewer #1: No

Reviewer #2: No

---

## [Author Response · Author response to Decision Letter 1]

10 May 2024

Thank you for your review and comments. Please see the attached response to reviewers letter for itemized revisions.

---

## [Decision Letter · Decision Letter 2]

28 May 2024

The prevalence of mental health and addiction concerns and factors associated with depression and anxiety during the COVID-19 pandemic in Ontario, Canada: A cross-sectional study

PONE-D-23-30089R2

Dear Dr. Markoulakis,

We’re pleased to inform you that your manuscript has been judged scientifically suitable for publication and will be formally accepted for publication once it meets all outstanding technical requirements.

Kind regards,

Guillermo Salinas-Escudero, PhD. MsC.

Academic Editor

PLOS ONE

Additional Editor Comments (optional):

Reviewers' comments:

Reviewer's Responses to Questions

**Comments to the Author**

1. If the authors have adequately addressed your comments raised in a previous round of review and you feel that this manuscript is now acceptable for publication, you may indicate that here to bypass the “Comments to the Author” section, enter your conflict of interest statement in the “Confidential to Editor” section, and submit your "Accept" recommendation.

Reviewer #1: All comments have been addressed

2. Is the manuscript technically sound, and do the data support the conclusions?

Reviewer #1: Yes

3. Has the statistical analysis been performed appropriately and rigorously? 

Reviewer #1: Yes

4. Have the authors made all data underlying the findings in their manuscript fully available?

Reviewer #1: Yes

5. Is the manuscript presented in an intelligible fashion and written in standard English?

Reviewer #1: Yes

6. Review Comments to the Author

Reviewer #1: I consider that the authors achieved a robust and interesting document ready to publish.

They carefully attended to the reviewers' observations and substantially improved the report.

The transparency of the document, the statistical analysis is more robust and the writing and discussion of results is more in-depth.

7. PLOS authors have the option to publish the peer review history of their article (what does this mean?). If published, this will include your full peer review and any attached files.

Reviewer #1: **Yes: **Castro-Ríos A

---

## [Editor Report · Acceptance letter]

3 Jun 2024

PONE-D-23-30089R2 

PLOS ONE

Dear Dr. Markoulakis, 

I'm pleased to inform you that your manuscript has been deemed suitable for publication in PLOS ONE. Congratulations! Your manuscript is now being handed over to our production team.

Kind regards, 

on behalf of

Dr. Guillermo Salinas-Escudero 

Academic Editor

PLOS ONE